# Ligand efficacy shifts a nuclear receptor conformational ensemble between transcriptionally active and repressive states

Brian S. MacTavish[1,11], Di Zhu[2,11], Jinsai Shang [1,3], Qianzhen Shao[4], Yuanjun He[2], Zhongyue J. Yang [4,5,6,7,8,9], Theodore M. Kamenecka[2] & Douglas J. Kojetin [1,2,5,6,7,10] ✉

Nuclear receptors (NRs) are thought to dynamically alternate between transcriptionally active and repressive conformations, which are stabilized upon ligand binding. Most NR ligand series exhibit limited bias, primarily consisting of transcriptionally active agonists or neutral antagonists, but not repressive inverse agonists—a limitation that restricts understanding of the functional NR conformational ensemble. Here, we report a NR ligand series for peroxisome proliferator-activated receptor gamma (PPARγ) that spans a pharmacological spectrum from repression (inverse agonism) to activation (agonism) where subtle structural modifications switch compound activity. While crystal structures provide snapshots of the fully repressive state, NMR spectroscopy and conformation-activity relationship analysis reveals that compounds within the series shift the PPARγ conformational ensemble between transcriptionally active and repressive conformations that are natively populated in the apo/ ligand-free ensemble. Our findings reveal a molecular framework for minimal chemical modifications that enhance PPARγ inverse agonism and elucidate their influence on the dynamic PPARγ conformational ensemble.

Nuclear receptors (NRs) are ligand-regulated transcription factors that control gene expression in response to binding endogenous metabolites and synthetic ligands, including about 15% of FDA-approved drugs[1]. NRs contain a modular domain architecture consisting of an N-terminal disordered activation domain, a central DNA-binding domain, and a C-terminal ligand-binding domain (LBD). A fundamental hypothesis in the NR field posits that in the absence of ligand the LBD exchanges between transcriptionally active and repressive states that can be stabilized upon ligand binding[2,3]. The binding of an agonist stabilizes an LBD conformation that favors the recruitment of coactivator proteins over corepressors, resulting in transcriptional activation of gene expression. Pharmacological antagonists block ligand binding and are generally considered transcriptionally neutral, stabilizing an LBD conformation that does not significantly alter coregulator recruitment compared to the unliganded state. Inverse agonists stabilize an LBD conformation that promotes interaction with corepressor proteins over coactivators, leading to transcriptional repression of gene expression. These ligand-dependent activities,

[1]Department of Integrative Structural and Computational Biology, Scripps Research and The Herbert Wertheim UF Scripps Institute for Biomedical Innovation & Technology, Jupiter, FL, USA. [2]Department of Molecular Medicine, Scripps Research and The Herbert Wertheim UF Scripps Institute for Biomedical Innovation & Technology, Jupiter, FL, USA. [3]Guangzhou Laboratory, School of Basic Medical Sciences, Guangzhou Medical University, Guangzhou, China. [4]Department of Chemistry, Vanderbilt University, Nashville, TN, USA. [5]Center for Structural Biology, Vanderbilt University, Nashville, TN, USA. [6]Vanderbilt Institute of Chemical Biology, Vanderbilt University, Nashville, TN, USA. [7]Center for Applied AI in Protein Dynamics, Vanderbilt University, Nashville, TN, USA. [8]Data Science Institute, Vanderbilt University, Nashville, TN, USA. [9]Department of Chemical and Biomolecular Engineering, Vanderbilt University, Nashville, TN, USA. [10]Department of Biochemistry, Vanderbilt University, Nashville, TN, USA. [11]These authors contributed equally: Brian S. MacTavish, Di Zhu. ✉e-mail: douglas.kojetin@vanderbilt.edu

which result in distinct coregulator recruitment and transcriptional outcomes, are known as ligand bias[4,5].

Although the field has a relatively good understanding of the structural basis of agonist-induced, coactivator-selective NR functions associated with transcriptional activation, several features remain poorly understood. These include graded transcriptional activity (e.g., partial vs. full agonism), transcriptional repression (inverse agonism), and whether agonists and inverse agonists select natively populated structural conformations (conformational selection) or induce non-native conformations. X-ray crystallography and cryo-electron microscopy have provided structural insights into DNA- and coregulator-bound NR transcriptional complexes, including LBDs bound to different pharmacological ligands[6,7]. However, these structural methods that capture static snapshots of functionally relevant NR complexes fail to explain features relevant to the dynamic functional LBD conformational ensemble, which are detectable by other methods such as NMR spectroscopy[8]. Moreover, studies into the mechanisms of graded NR transcriptional repression have been hampered because most NR ligand series consist of compounds of a single pharmacological type (e.g., agonists). Less common are reports of transcriptionally repressive NR inverse agonists and compounds that span the full pharmacological spectrum, from agonism to inverse agonism, which display distinct coregulator recruitment profiles or ligand bias.

Peroxisome proliferator-activated receptor gamma (PPARγ) is a lipid-sensing NR and target of antidiabetic drugs that function as pharmacological agonists. NMR studies revealed that in the absence of a ligand, the apo-PPARγ LBD is a dynamic conformational ensemble and samples two or more structural conformations, which can be stabilized into a single active state upon binding an agonist[9]. Hydrogen/deuterium exchange mass spectrometry revealed that the binding of graded agonists differentially stabilizes a critical structural element, helix 12, that is critical for NR activation[10] supporting a role for protein dynamics in the mechanism of graded PPARγ agonism[11]. Crystal structures of PPARγ LBD bound to graded agonists typically show the same transcriptionally active LBD conformation[12]. In contrast, functional profiling of graded agonists binding to the PPARγ in biochemical assays using purified LBD correlates well to graded transcriptional activation of full-length PPARγ in cells and NMR-detected shifts of the PPARγ LBD conformational ensemble from a ground state towards an active state[13,14].

The PPARγ inverse agonist, T0070907, was originally described as an antagonist[15] because it and a chemically related compound, GW9662[16], bind covalently through a nucleophilic substitution mechanism to Cys285 within the LBD and blocked binding of other PPARγ ligands. However, subsequent studies showed these compounds do not block all ligands from binding PPARγ[17–21]. Separate from their potential functions in blocking ligand binding, these compounds have distinct pharmacological properties. GW9662 is a transcriptionally neutral PPARγ ligand, whereas T0070907 is an inverse agonist that represses PPARγ transcription[22]. T0070907 and GW9662 share the same 2-chloro-5-nitrobenzamide scaffold, and substitutions at the amide $R_1$ group can be modified to elicit PPARγ agonism[23] or inverse agonism[22,24–27]. Covalent PPARγ inverse agonists are currently being developed as potential therapeutics in bladder cancer where hyperactivation of PPARγ transcription occurs[28–31]. Crystal structures of PPARγ LBD bound to corepressor peptide and inverse agonists analogs of T0070907[24–27] have provided insight into low-energy structural snapshots of PPARγ forced into a fully repressive state by the peptide. However, these static structures do not inform on the structural mechanism, or conformation-activity relationship, behind the differential graded activity observed within a ligand series during structure-activity relationship development, as they do not elucidate how compounds with graded activity influence the dynamic PPARγ LBD conformational ensemble.

Here, we show that the 2-chloro-5-nitrobenzamide scaffold offers an opportunity to understand how ligands with graded activity across the entire pharmacological spectrum influence the PPARγ LBD conformational ensemble. We found that relatively minimal chemical modifications to this scaffold relative to the parent compound T0070907 yielded a ligand series spanning graded inverse agonism to agonism. Crystal structures of PPARγ LBD bound with ligands and corepressor peptide along with density functional theory (DFT) calculations provide structural insight into the structural basis of improved inverse agonist efficacy. NMR studies reveal that compounds within the ligand series influence the function of PPARγ by shifting the LBD conformational ensemble between transcriptionally active- and repressive-like states that are natively populated in the apo-LBD ensemble. Correlation analysis shows that biochemical activities and NMR structural analysis using purified PPARγ LBD protein can explain, and perhaps predict, ligand-dependent functions of full-length PPARγ-mediated transcription and gene expression in cells.

## Results

### Design hypothesis to improve inverse agonism

Crystal structures of PPARγ LBD in transcriptionally active and repressive conformations suggest features that may be important for inverse agonism imparted by T0070907[24]. In the transcriptionally active conformation, where PPARγ LBD is bound to agonist (e.g., rosiglitazone) and coactivator (e.g., TRAP220/MED1) peptide, helix 12 is solvent exposed and forms the AF-2 surface along with helix 3–5 for LXXLL-containing motifs in coactivator proteins (Fig. 1a). In contrast, in the repressive conformation helix 12 occupies the orthosteric ligand-binding pocket leaving the remaining AF-2 surface regions exposed for binding the longer corepressor peptide motif. Furthermore, in the repressive conformation a pi-stacking interaction is observed between the polar T0070907 pyridyl group and three residues (His323, His449, and Tyr473) forming an aromatic triad that includes a water-bridged interaction between the pyridyl nitrogen and the His323 imidazole ring (Fig. 1b).

NMR studies also inform the structural mechanism of PPARγ inverse agonism. In the absence of ligand, helix 12 exchanges in and out of the orthosteric ligand-binding pocket in repressive- and active-like conformations, respectively, on the microsecond-to-millisecond time scale[24]. T0070907 binding slows the rate of exchange between these two natively populated LBD conformations such that they are long-lived and simultaneously observed by NMR[22,32,33]. The active-like T0070907-bound state binds coactivator peptides with high affinity, and the repressive-like state binds corepressor peptides with high affinity[22]. Binding of GW9662, which contains a hydrophobic phenyl $R_1$ moiety instead of the polar pyridyl group in T0070907, also exchanges between these two conformations, but primarily populates an active-like conformation that is forced into a repressive conformation upon binding corepressor peptide[22,33].

Taken together, these findings informed a hypothesis that 2-chloro-5-nitrobenzamide compounds with polar aromatic $R_1$ groups may stabilize the transcriptional repressive conformation via pi-stacking of the $R_1$ group with the aromatic triad (Fig. 1c). To test this hypothesis, we initially searched the ZINC database[34] and discovered a commercially available compound (ZINC5672437) with a 4-carbamoylphenyl group—a polar benzamide extended from an otherwise non-polar aromatic phenyl ring. ZINC5672437 displays partial corepressor-selective inverse agonism relative to GW9662 and T0070907 in a time-resolved fluorescence resonance energy transfer (TR-FRET) corepressor peptide interaction assay (Fig. 1d) and a cell-based transcriptional reporter assay (Fig. 1e).

T0070907 displays flipped ligand-binding modes where the pyridyl ring points towards the β-sheet and AF-2 surface in the active and repressive conformations, respectively, when comparing crystal structures of PPARγ LBD T0070907 is soaked into preformed apo-PPARγ LBD crystals (active) or cobound with NCoR1 ID2 corepressor

peptide (repressive)[19,22,24]. Paramagnetic relaxation enhancement (PRE) NMR studies showed the T0070907- and GW9662-bound PPARγ LBD exchanges between these two conformations[24]. To determine if GW9662 and ZINC5672437 can adopt flipped ligand-binding modes, we solved two repressive conformation crystal structures of PPARγ LBD bound to NCoR1 ID2 peptide-bound crystal structures of PPARγ LBD bound to GW9662 or ZINC5672437 each to 1.8 Å resolution (Supplementary Table 1 and Supplementary Fig. 1). We also solved an active conformation crystal structure of ZINC5672437-bound PPARγ LBD to 2.1 Å resolution by ligand soaking into preformed apo-PPARγ

LBD crystals (Supplementary Table 1 and Supplementary Fig. 1) for comparison to previously determined active conformation structures bound to GW9662 and T0070907 that were similarly obtained by ligand soaking[19,22]. The crystal structures show that all three compounds are capable of adopting the active- and repressive-like ligand conformations (Fig. 1f). The 4-carbamoylphenyl group extending from the non-polar aromatic ring in ZINC5672437 replaces the water-bridged T0070907 pyridyl interaction with His323 and pushes the bridging water towards the AF-2 surface (Fig. 1g). 2D [¹H,¹⁵N]-TROSY-HSQC NMR focused on Gly399, a residue located near the AF-2

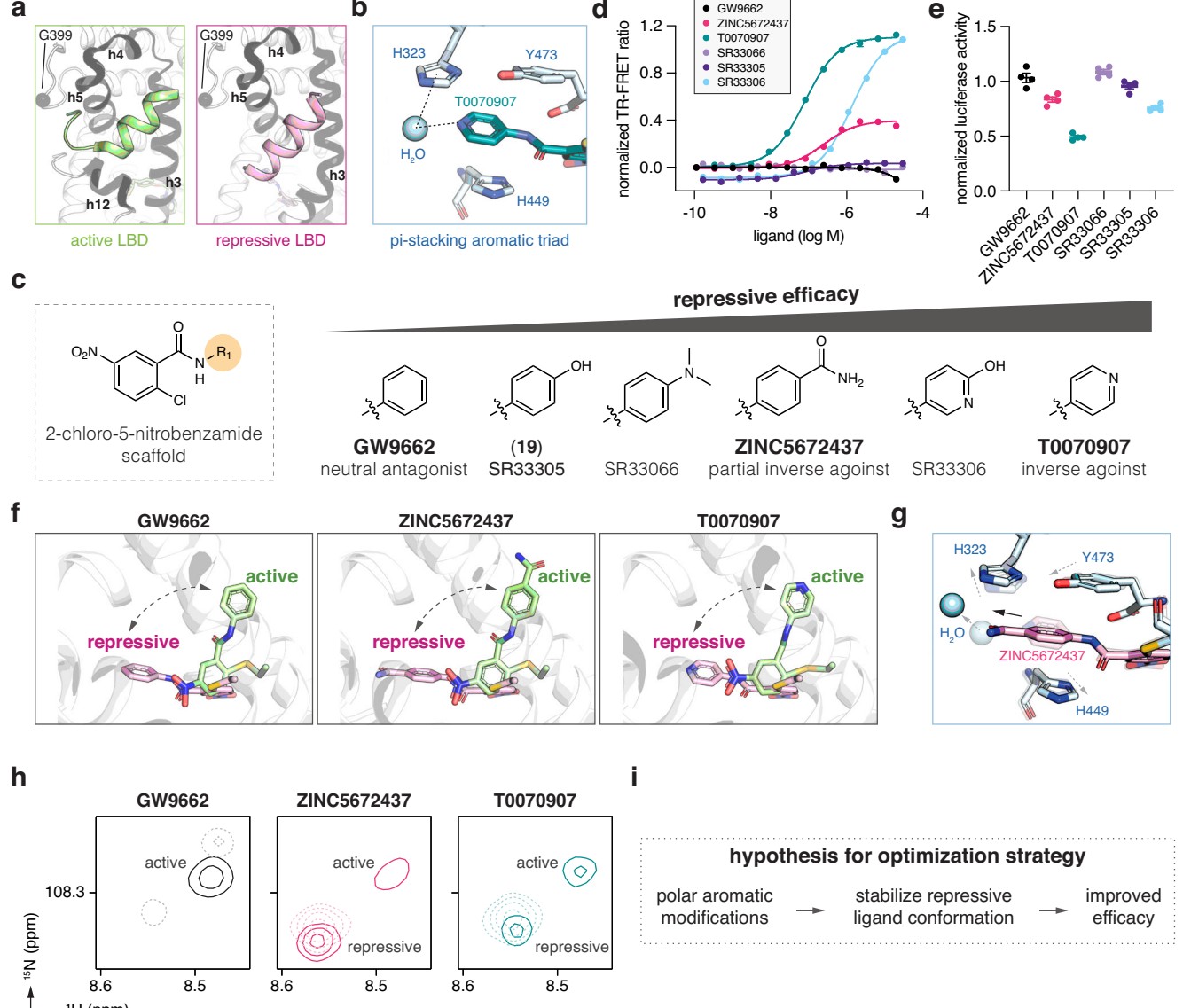

**Fig. 1 | Structure-function data inform a design hypothesis to improve PPARγ inverse agonism. a** AF-2 surface differences between PPARγ LBD in the transcriptionally active conformation bound to agonist rosiglitazone and TRAP220/MED1 coactivator peptide (PDB 6ONJ) and repressive conformation bound to inverse agonist T0070907 and NCoR1 corepressor peptide (PDB 6ONI) peptides. Coactivator and corepressor peptides are shown in green and pink, respectively. Helical structural elements are noted (h3, h4, h5, h12) and residue names are called out. **b** Pi stacking between the polar T0070907 pyridyl group and three residues (His323, His449, and Tyr473) and a bridging water molecule form in the transcriptionally repressive conformation (PDB 6ONI). **c** Compound scaffold of parent molecules containing polar pyridyl (T0070907) and hydrophobic phenyl (GW9662) groups in relation to the polar benzamide group in ZINC5672437. **d** TR-

FRET NCoR1 corepressor biochemical interaction assay (n = 3 biological replicates; mean ± s.e.m.). **e** Cellular luciferase transcriptional reporter assay in HEK293T cells treated with 10 μM compound (n = 4 biological replicates; mean ± s.e.m.). **f** Crystal structures showing the compound flipped binding modes in transcriptionally active vs. repressive conformations when bound to GW9662 (PDB 3B0R and 8FHE), T0070907 (PDB 6C1I and 6ONI), or ZINC5672437 (PDB 8FHF and 8FHE). **g** ZINC5672437 pi-stacking interaction in the repressive state (PDB 8FHE) where the benzamide replaces the bridging water in the T0070907-bound structure (PDB 6ONI). **h** 2D [¹H,¹⁵N]-TROSY-HSQC NMR focused on Gly399 of ¹⁵N-labeled PPARγ LBD bound to GW9662, T0070907, or ZINC5672437 in the absence (solid lines) or presence (dashed lines) of NCoR1 corepressor peptide. **i** Data-informed design hypothesis to improve PPARγ inverse agonism.

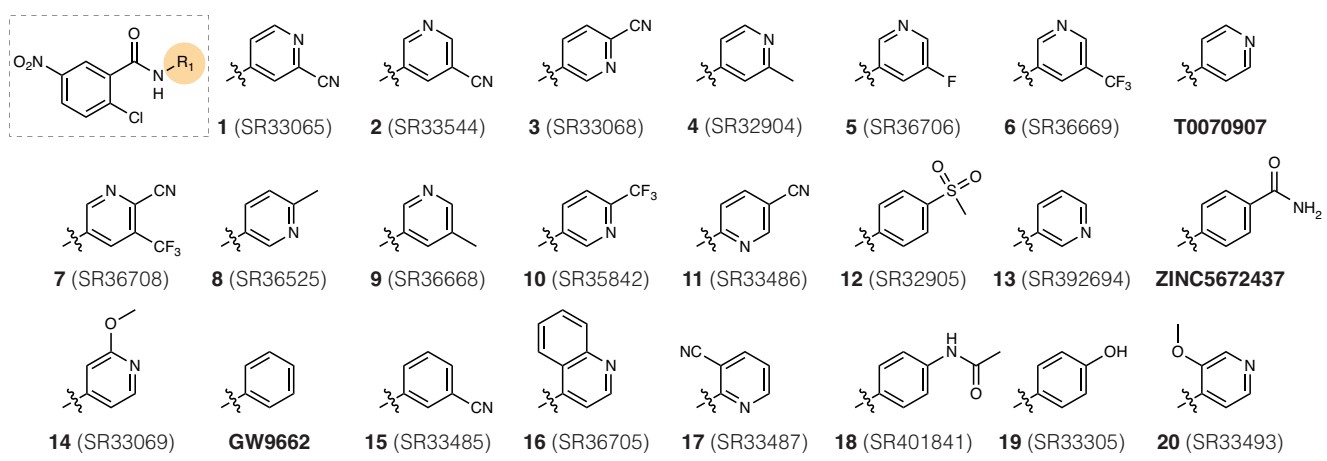

**Fig. 2 | Compound analogs synthesized in this study.** Compounds are numbered via the rank order in efficacy in the TR-FRET NCoR1 corepressor peptide interaction assay data shown in Fig. 3.

coregulator interaction surface (Fig. 1a) that is sensitive to ligand-dependent active and repressive conformations[13,22], reveals the presence of two long-lived ZINC5672437-bound LBD conformations or structural populations (Fig. 1h) that are similar to the T0070907-bound active- and repressive-like conformations[22]. The addition of NCoR1 ID2 corepressor peptide consolidated these two structural populations into a single conformation. In contrast, GW9662 populates an active-like LBD conformation in the absence of peptide and two distinct conformations when bound to NCoR1 peptide[22].

The biochemical, cellular, and NMR data show that ZINC5672437 functions as a partial PPARγ inverse agonist with graded activity between T0070907 and GW9662 via a polar substation in ZINC5672437 that displaces or replaces the water-bridged interaction between the pyridyl nitrogen and the His323 imidazole ring in T0070907. To determine if other polar substitutions in the 4-position of GW9662 could substitute for the water-bridged interaction, we synthesized 4-(dimethylamino)phenyl (SR33066) and 4-hydroxyphenyl (SR33305, **16**) analogs and tested them in the TR-FRET (Fig. 1d) and cell-based transcriptional reporter assay (Fig. 1e). Relative to GW9662, SR33066 did not improve repressive activity and SR33305 only showed a slight improvement. In contrast, appending a 4-hydroxy modification onto a pyridyl ring (SR33066; 6-hydroxypyridin-3-y) increased repressive activity relative to SR33055 (**19**). However, SR33066 shows reduced transcriptional efficacy relative to T0070907, which may result from decreased potency (i.e., covalent reactivity) that was observed in the TR-FRET assay possibly from unfavorable interactions in its pre-covalent ligand encounter complex binding pose[27]. Taken together, these data provide support for an aromatic triad-stabilizing hypothesis and indicate that chemical modifications containing aromatic polarity may stabilize a repressive PPARγ LBD conformation and improve transcriptional repression efficacy (Fig. 1i).

### Ligand series that spans graded repression and activation

We previously showed that coactivator-selective graded transcriptional activation of PPARγ occurs via shifting the conformational ensemble of the PPARγ LBD from a ground state towards a fully active state[13]. We sought to determine if corepressor-selective inverse agonism may function through a related mechanism. Towards this goal, we characterized twenty 2-chloro-5-nitrobenzamide analogs containing an aromatic R1 group decorated with different substituents to determine if we can improve efficacy and generate a ligand set with graded repressive activity with minimal chemical modifications (Fig. 2). We tested the compounds in several biochemical and cellular ligand profiling assays that report on active and repressive PPARγ functions as we did in our study of graded PPARγ agonism[13].

Using TR-FRET biochemical assays, we determined how the analogs affect the interaction between the PPARγ LBD and coregulator peptides derived from the NCoR1 corepressor and MED1 coactivator proteins (Fig. 3a). In these experiments, the assay values report on the relative change in affinity or interaction efficacy between PPARγ LBD and the coregulator peptides; larger values correspond to increased interaction efficacy. To directly measure how the analogs affect coregulator peptide binding affinity to PPARγ LBD, we performed fluorescence polarization (FP) assays using NCoR1 corepressor and TRAP220/MED1 coactivator peptides (Fig. 3b). We also calculated a coregulator bias factor for the TR-FRET and FP data, which reports on the difference of NCoR1 and MED1 TR-FRET efficacy and FP determined affinity (Fig. 3c). To determine how the analogs affect PPARγ transcription, we performed a luciferase reporter assay where HEK293T cells were transfected with a full-length PPARγ expression plasmid along with a second plasmid containing three copies of the PPAR DNA-binding response element sequence (PPRE) upstream of luciferase gene and treated the cells with compounds (Fig. 3d). Finally, to determine how the analogs affect PPARγ-mediated gene expression, we cultured 3T3-L1 preadipocytes in the presence of compounds and measured the expression of the adipogenic PPARγ target gene *aP2/FABP4* after 2 days of differentiation (Fig. 3e).

Overall, the compound analogs have a wide range of graded activities spanning transcriptional repression, with increased corepressor recruitment and binding affinity, to transcriptional activation, with increased coactivator recruitment and binding affinity. Approximately half of the compounds display similar or improved inverse agonism compared to T0070907; of these, compounds **3** (SR33068) and **7** (SR36708) are the most efficacious in repressing *aP2* expression in 3T3-L1 cells to levels similar to undifferentiated cells. Several analogs show properties of PPARγ agonism via increased coactivator peptide interaction, decreased corepressor peptide interaction, and increased transcription and *aP2* expression, including compounds **17** (SR33487), **18** (SR401841), **19** (SR33305), and **20** (SR33493). However, compared to the orthosteric full agonist rosiglitazone, the activity of compounds 17-20 classify these analogs as partial agonists.

Some general features of chemical modifications that confer inverse agonism or partial agonism are apparent. For example, all the improved inverse agonists contain a polar pyridyl aromatic ring, most of which are appended with a polar substitution (cyano, fluoro, or trifluoromethyl). In contrast, analogs with activities ranked worse than GW9662—the least efficacious inverse agonists and partial agonists—all contain a hydrophobic non-polar phenyl ring, with two exceptions. Compounds **16** (SR36705) and **17** (SR33487) have extensions off a similar position of a polar pyridyl ring, isoquinolin-1-yl, and 3-

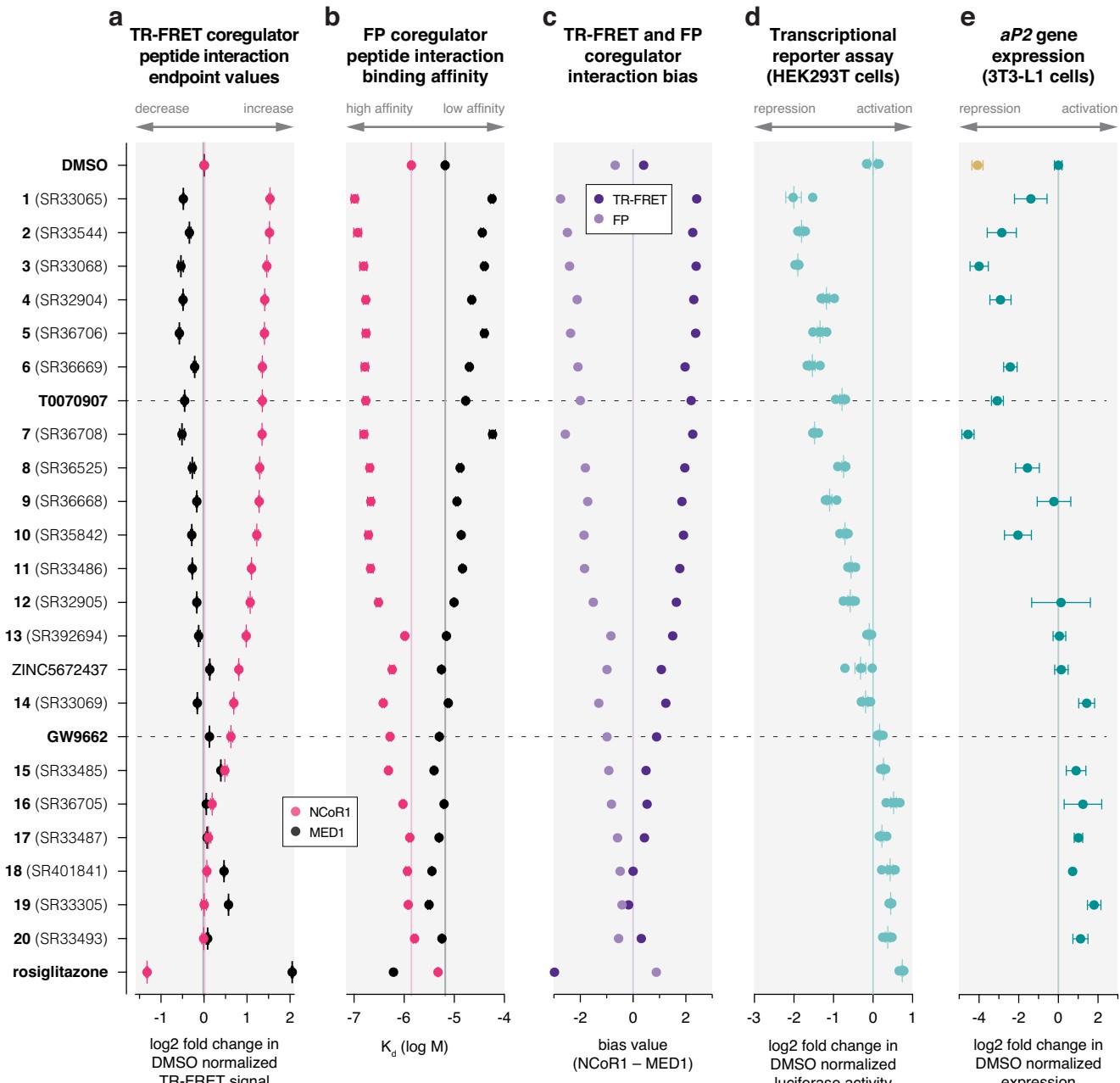

**Fig. 3 | Compounds show graded activity in biochemical and cellular functional profiling assays. a** Time-resolved fluorescence resonance energy transfer (TR-FRET) NCoR1 corepressor peptide biochemical interaction assay ($n = 3$ biological replicates; mean ± s.e.m.). **b** Fluorescence polarization (FP) NCoR1 corepressor peptide binding assay ($n = 3$ biological replicates) fitted affinity (mean ± std. err.). **c** Coregulator bias values in the TR-FRET and FP assay data were calculated by subtracting the MED1 value from the NCoR1 value. **d** Cellular luciferase transcriptional reporter assay in HEK293T cells ($n = 4$ biological replicates; mean ± s.e.m.). **e** Expression of *aP2* in 3T3-L1 cells after 2 days of differentiation ($n = 3$ biological replicates; mean ± upper and lower limits); tan circles represent undifferentiated cells. For all panels, vertical lines when present represent DMSO control values and dashed horizontal lines note parent compounds.

cyanopyridin-2-yl, respectively, indicating extensions at these positions may be unfavorable for inverse agonism. However, SAR on a larger ligand series would be needed to confirm these observations.

In our previous study of thiazolidinedione (TZD) PPARγ agonists, we found that graded activation or efficacy within the TZD ligand series via increased coactivator peptide recruitment and binding affinity is highly correlated to cellular transcription[13]. Spearman correlation coefficients calculated from pairwise comparisons of the 2-chloro-5-nitrobenzamide biochemical and cellular profiling assay data reveal strong efficacy correlations among the biochemical and cellular data (Fig. 4 and Supplementary Fig. 2). Transcriptional repression and

decreased *aP2* expression are highly correlated to the NCoR1 TR-FRET (increased NCoR1 interaction) and FP (stronger NCoR1 binding affinity) biochemical assay data and highly anti-correlated to the MED1 TR-FRET (decreased MED1 interaction) and FP (weakened MED1 binding affinity) biochemical assay data. Coregulator bias values calculated from the TR-FRET and FP data are also highly correlated to cellular functions. However, biochemical and cellular ligand potency shows a poor correlation to ligand efficacy for the 2-chloro-5-nitrobenzamide ligand series (Supplementary Fig. 3), likely because these compounds bind via a covalent mechanism, unlike the TZD ligand series that bind via a non-covalent mechanism where ligand-binding kinetics contributes to

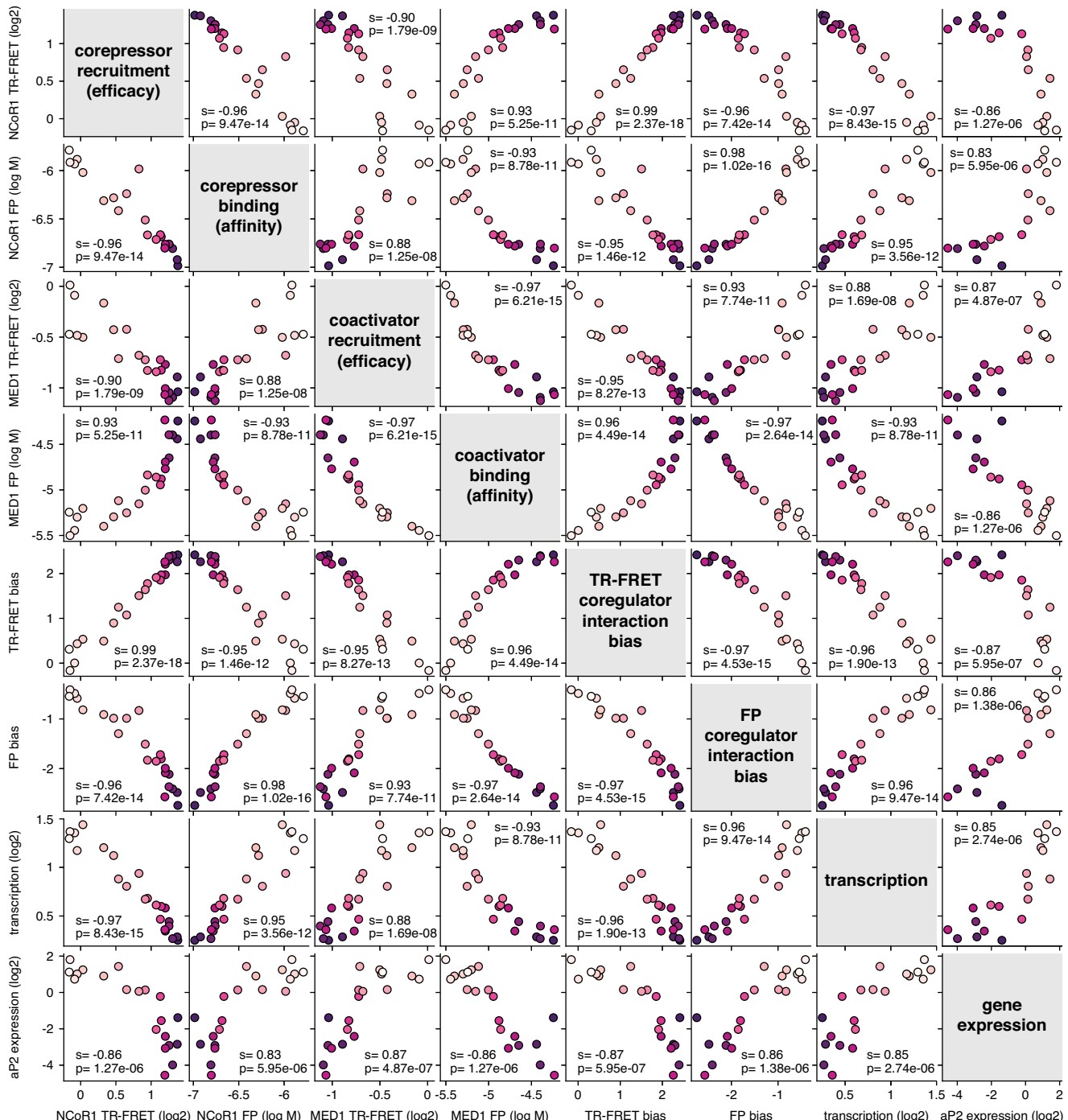

**Fig. 4 | Pairwise correlation analysis of the compound profiling assay data.** Spearman correlation coefficients and associated two-sided *p* values are listed. Data (from Fig. 3) are colored from white to purple according to the compound numbers displayed in Fig. 2 (i.e., ranking of functional efficacy in the NCoR1 TR-FRET assay from Fig. 3a). A matrix of Spearman correlation coefficients is also shown in Supplementary Fig. 2.

ligand efficacy[13]. These data show that functional efficacy profiling of the 2-chloro-5-nitrobenzamide ligand series in 384-well microplate biochemical assays using purified PPARγ LBD protein explains the effect of the ligand series on the transcriptional activity of full-length PPARγ and the expression of a PPARγ target gene during adipogenesis.

## Crystal structures inform on improved corepressor-selective efficacy

To determine the structural basis of improved inverse agonism, we solved crystal structures of PPARγ LBD bound to NCoR1 corepressor peptide and compounds 2 (SR33544), 3 (SR33068), 4 (SR32904), 5

(SR36706), and 11 (SR33486) with resolutions ranging from 1.42 to 2.22 Å (Supplementary Table 1 and Supplementary Fig. 1). Overall, the repressive PPARγ LBD conformation in these structures (Fig. 5a) is highly similar to crystal structures of PPARγ LBD bound to NCoR1 corepressor peptide and T0070907[24], GW9662, and ZINC5672437 (0.15 Å Cα rmsd). Helix 12 adopts a solvent-occluded conformation within the orthosteric ligand-binding pocket, which exposes a larger AF-2 surface that enables the binding of the NCoR1 CoRNR motif. The corepressor CoRNR motif[35] is ~1 helical turn longer than the coactivator LXXLL motif that requires and interacts with a solvent-exposed helix 12 conformation[36].

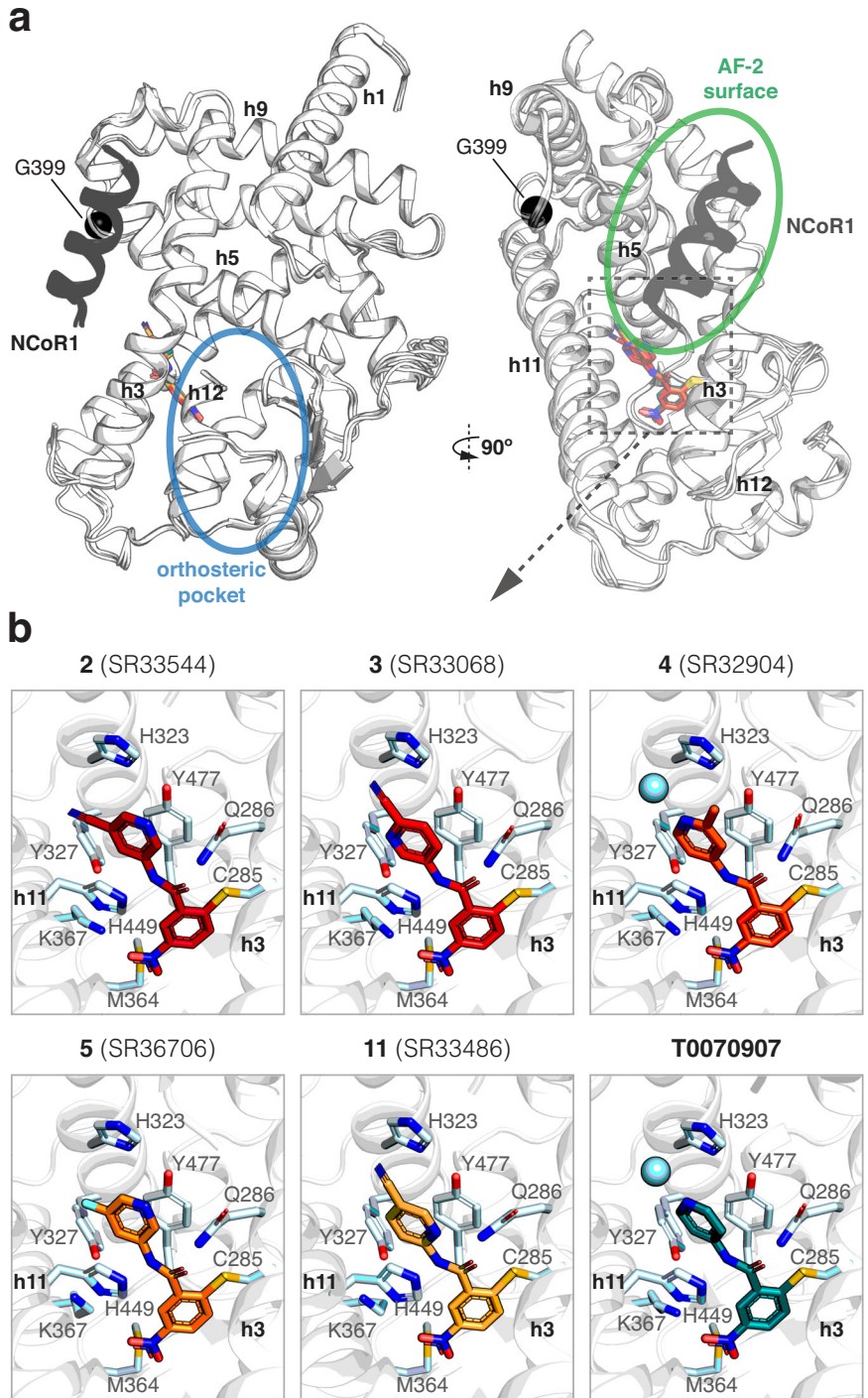

**Fig. 5 | Inverse agonist binding modes in co-crystal structures of PPARγ LBD cobound to NCoR1 corepressor peptide. a** Structural overlay of PPARγ LBD bound to NCoR1 peptide and SR33544 (**2**, PDB 8FKC), SR33068 (**3**, PDB 8FKD), SR32904 (**4**, PDB 8FKE), SR36706 (**5**, PDB 8FKF), SR33486 (**11**, PDB 8FKG), and T0070907 (PDB 6ONI). Annotations show the location of the orthosteric ligand-binding pocket, AF-2 coregulator interaction surface, and Gly399 that serves as a structural proxy for AF-2 surface conformation in the NMR analysis in Fig. 6. **b** Structures are oriented with a view where the ligand R₁ group is pointed towards

the AF-2 surface. Residues annotated were considered in density functional theory (DFT) quantum mechanical (QM) calculations of ligand interaction free energies (ΔG_bind), and α-helices are noted by h3 (helix 3) and h11 (helix 11). Compounds are colored red to orange according to the compound numbers (functional efficacy in the NCoR1 TR-FRET assay) displayed in Fig. 2; the parent compound T0070907 is colored teal. Helical structural elements are noted (h3, h4, h5, h12) and residue names are called out.

All of the crystallized compounds contain a pyridyl nitrogen, which in T0070907 is at the fourth position, with additional R₁ groups that interact with the aromatic triad residues (His323, His449, Tyr473) via pi-stacking interactions (Fig. 5b). Compounds 3 and 11 contain a cyano group at the fourth position with a pyridyl nitrogen at the third

and second positions, respectively, which substitutes for the water that bridges the interaction of T0070907 with the side chain of His323. Compound 2 contains a pyridyl nitrogen at the third position pointing towards the NCoR1 corepressor peptide bound at the AF-2 surface, and a cyano group at the fifth position that points towards helix 11 with no

water-bridged interaction with the side chain of His323. Compound 5 contains a pyridyl nitrogen at the third position that points towards the NCoR1 corepressor peptide, similar to compound 2, but contains a fluorine at the fifth position instead of a cyano group that points towards helix 11. Finally, compound 4 contains a pyridyl nitrogen at the fourth position similar to T0070907, which retains the water-bridged interaction with His232, and a methyl substitution at the third position that points towards the NCoR1 corepressor peptide.

We performed DFT quantum mechanics (QM) calculations to estimate the interaction free energies ($\Delta G_{bind}$) between the compounds with residues comprising the pi-stacking aromatic triad residues (His323, His449, Tyr473) and others nearby residues (Cys285, Gln286, Tyr327, Met364, Lys367) that make contacts to, or are near, the compound in the crystal structures described herein. For two compounds, 3 and 11, we also modeled an $R_1$ ring conformation flipped by 180° and performed DFT-QM calculation, as the electron density could not differentiate the final modeled or flipped orientations. Compared to T0070907, the substituted pyridyl $R_1$ groups in compounds 2, 3, 5, and 11 afford more favorable $\Delta G_{bind}$ values (Supplementary Table 2). This suggests that the improved compound functional efficacy may originate in part from additional aromatic ring polarity and polar compound modifications projecting towards helix 11 or the bound corepressor peptide. However, among the pyridyl-substituted analogs, $\Delta G_{bind}$ is most favorable for compound 11, which displays the lowest efficacy among this group. Thus, there may be additional contributions from other residues, water-bridging interactions, or other phenomena not considered in the DFT-QM calculations such as compound structural rigidity that may contribute to experimental ligand efficacy.

### Ligands shift the functional LBD conformational ensemble

We posited that the improved inverse agonism within the ligand series could originate from two effects on the PPARγ LBD conformational ensemble: the ligands could either maintain the two long-lived repressive and active-like LBD conformations observed for T0070907-bound LBD[22] and further slow the rate of exchange between these states, or alternatively, the ligands could shift the ensemble towards and stabilize only the repressive-like conformation, while agonists would shift the ensemble towards an active-like conformation. To probe these mechanisms, we compared 2D [$^1$H,$^{15}$N]-TROSY-HSQC NMR data of $^{15}$N-labeled PPARγ LBD bound to the ligand series.

Focusing on Gly399 located near the AF-2 surface (Figs. 1a and 5a), compounds with improved inverse agonism shift the LBD conformational ensemble towards a single peak representative of the repressive-like conformation (Fig. 6a). Other improved analogs or compounds with partial inverse agonism show two NMR peaks, a repressive-like conformation peak and a weakly populated peak that is on-path (along the diagonal relative to the two T0070907-observed states) to the active-like conformation and in slow exchange on the NMR time scale (milliseconds or longer). Analogs within the ligand series that function as PPARγ agonists, compounds 15–20, stabilize either a single active-like conformation peak or show two NMR peaks that are shifted towards a graded agonism state similar to the graded TZD agonist series[13]. In addition to Gly399, other residues throughout the LBD with well-resolved NMR peaks show ligand-activity-dependent NMR peak shifts between repressive- and active-like states (Supplementary Figs. 4–9). We observed similar shifts for these residues in our previous study on T0070907[22], including Arg234 (helix 2), Gly338 (β-sheet in the ligand-binding pocket), Arg350 (helix 6), Asn375 (helix 7), and Asp380 (helix 8-9 loop). This indicates the entire LBD is sensitive to the ligand-induced graded shift in the conformational ensemble between repressive- and active-like states.

To quantitatively determine the influence of the ligand series on the PPARγ LBD conformational ensemble, we calculated population-weighted $^1$H NMR chemical shift values for Gly399 in $^{15}$N-labeled PPARγ LBD when bound to the compounds. This measurement reports on the degree to which the compounds shift the LBD ensemble between repressive- and active-like states. Spearman correlation coefficients calculated from pairwise comparisons of the population-weighted NMR chemical shifts and biochemical and cellular ligand profiling assays show a strong correlation (Fig. 6b and Supplementary Fig. 2). However, there are limitations to this analysis, particularly the assumption that the populations of distinct, long-lived, ligand-bound LBD conformations are linked to the measured integrated volumes of well-resolved NMR peaks. This assumption may not be accurate, as dynamic motions on different time scales can influence NMR peak line shapes and thus integrated peak volumes. Furthermore, in some cases, there may be overlapped peaks that cannot be easily distinguished and picked separately for peak volume integration. Notwithstanding these limitations, the analysis indicates that the NMR-detected shift in the structural populations explains the graded activity within the ligand series with relatively high correlation Spearman coefficients (>0.7), supporting the underlying hypothesis that compounds within the ligand series shift the PPARγ LBD conformational ensemble between active and repressive conformations.

## Discussion

Structural biology studies of NRs focused on understanding the molecular basis for ligand-dependent changes in transcription have been primarily focused on generating crystal structures of NR LBDs bound to coregulator peptide and/or ligand. However, while crystal structures provide low-energy static snapshots of the functional endpoints (e.g., fully active states) they do not explain graded activity that is critical for understanding how ligands influence the dynamic NR LBD conformational ensemble. PPARγ is a good model system to study how pharmacological ligands influence the dynamic NR LBD conformational ensemble. Although crystal structures of PPARγ LBD bound to partial or full agonists all show a similar low-energy active LBD conformation and do not explain the molecular basis of graded agonism[11,14], NMR studies revealed that graded activation of PPARγ is correlated to shifting the LBD conformational ensemble from a ground state to an active state[13]. Here, we define a more comprehensive LBD conformational ensemble and show that compounds within the same scaffold select natively populated repressive- and active-like conformations present in the apo/ligand-free state[24].

The ligand series we developed, based on the T0070907 2-chloro-5-nitrobenzamide compound scaffold, displays graded activity that selects LBD conformations detected by NMR along the transcriptionally repressive-to-active continuum that are natively populated active and repressive LBD conformations. Other reported PPARγ inverse agonists appear to function via a different mechanism that does not select for a natively populated repressive state. Instead, crystal structures of PPARγ LBD bound to these compounds show solvent-exposed non-active helix 12 conformations with ligand-binding modes that overlap with the repressive helix 12 conformations within the orthosteric pocket[37,38]. These compounds may bind to the LBD similar to TZDs, which are orthosteric agonists that bind via a two-step induced fit mechanism with an initial encounter complex followed by a transition into the orthosteric pocket, which displaces helix 12 from its apo-conformation within the orthosteric pocket[39]. Thus, there appear to be two different mechanisms of PPARγ inverse agonism. Inverse agonists including T0070907 and analogs generated here select a native-like repressive conformation where helix 12 adopts a conformation within the orthosteric ligand-binding pocket that we showed previously[22,24] and here results in high-affinity corepressor peptide interaction. Alternatively, orthosteric inverse ligands that bind to the orthosteric pocket via an induced fit mechanism competing with and displacing helix 12 from the orthosteric pocket appear to function

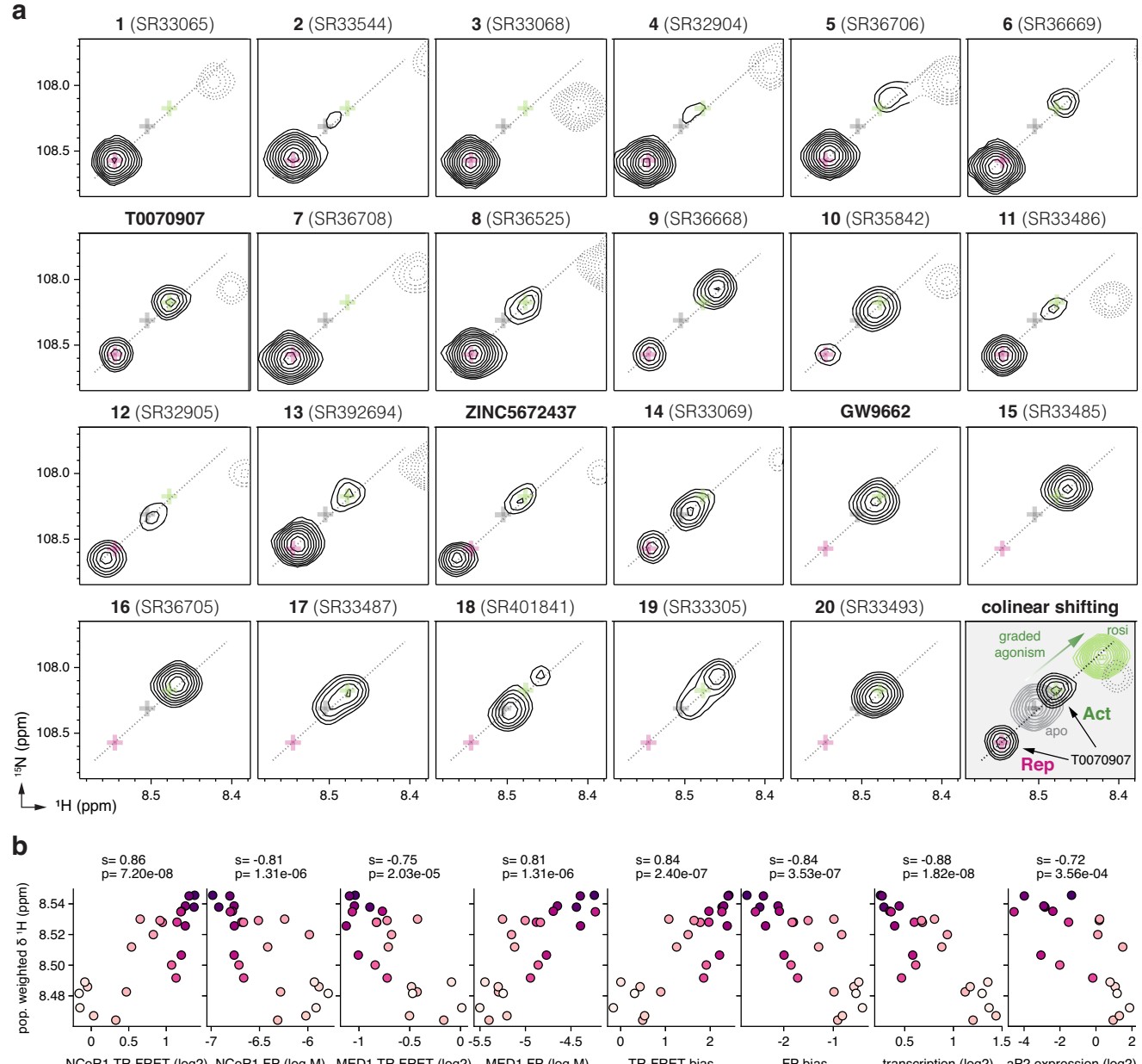

**Fig. 6 | NMR analysis PPARγ LBD conformational ensemble describes biochemical and cellular functions of the ligand series. a** 2D [$^1$H,$^{15}$N]-TROSY-HSQC NMR of $^{15}$N-labeled PPARγ LBD bound to compounds in the ligand series focused on Gly399, a residue located near but not within the AF-2 surface. The bottom right panel illustrates: peak positions of the active-like (Act) and repressive-like (Rep) states populated by T0070907 (two black peaks noted with green and pink plus signs, respectively); the peak position of the apo/ligand-free state (gray peak and gray plus sign); a dotted line noting the colinear shifting between active and repressive states; and the direction peaks shift for graded (partial-to-full) agonism

is indicated with a white-to-green arrow that points to the rosiglitazone (full agonist)-bound state (green peak). The dotted NMR peak corresponds to residue Gly284 that is stabilized in this region of the 2D NMR spectrum by repressive compounds in the ligand series. **b** Pairwise correlation plots between compound profiling data (from Fig. 3) and weighted NMR chemical shift population shifts. Spearman correlation coefficients and associated two-sided *p* values are listed. Data are colored from white to purple according to the compound numbers displayed in Fig. 2 (i.e., ranking of functional efficacy in the NCoR1 TR-FRET assay from Fig. 3a). A matrix of Spearman correlation coefficients is also shown in Supplementary Fig. 2.

primarily by coregulator inhibition: weakening coactivator peptide interaction while not significantly changing or even slightly inhibiting corepressor peptide interaction[22].

Our work here shows that NMR enables conformation-activity relationship analysis and explains activities for a ligand series spanning transcriptional activation and repression. In contrast, crystal structures coupled to SAR studies only reveal the fully repressive LBD conformation and do not reveal the structural mechanism of graded repression. Recent studies have also reported improved covalent PPARγ inverse agonists[25–27] containing more elaborate R$_1$ group

chemical modifications in contrast to the relatively minimal chemical modifications that we show here are sufficient to push the LBD conformational ensemble towards a fully repressive state. Interestingly, compounds in one of these series contain different scaffold structures, and crystal structures revealed an LBD conformation similar to the orthosteric inverse agonist-bound conformation[37,38] indicating these analogs likely function via coregulator inhibition similar to orthosteric inverse agonists instead of corepressor-selectivity. With the growing interest in developing PPARγ inverse agonists for bladder cancer therapeutics, including a covalent analog that is currently in phase 1

clinical trials[40,41], our findings demonstrate a platform that can assess, explain, and potentially predict the cellular transcriptional activity of PPARγ compounds ranging from agonism to inverse agonism via biochemical and NMR-detected structural biology studies focused on the PPARγ LBD.

## Methods

### Ligands and peptides
Commercially available compounds, T0070907 (CAS 313516-66-4) and GW9662 (CAS 22978-25-2), were purchased from Cayman Chemical. Details for the compounds synthesized in this study can be found in Supplementary Methods. Peptides derived from human NCoR1 ID2 (2256-2278; DPASNLGLEDIIRKALMGSFDDK) and human TRAP220/MED1 ID2 (residues 638–656; NTKNHPMLM NLLKDNPAQD) were synthesized by LifeTein with an amidated C-terminus for stability, with or without a N-terminal FITC label and a six-carbon linker (Ahx). Ligand concentrations for final functional profiling assays described below were chosen based on the observed potency values in biochemical TR-FRET coregulator and cellular transcriptional reporter assays after ~2 h ligand treatment (see Source Data 1). Because 2-chloro-5-nitrobenzamide compound analogs bind via a covalent mechanism, their potency values will progressively left shift (become more potent) with time.

### Cell lines for mammalian cell culture
HEK293T (ATCC #CRL-11268) and 3T3-1L (ATCC #CL-173) cells were cultured according to ATCC guidelines. HEK293T cells were grown at 37 °C and 5% $CO_2$ in Dulbecco's Modified Eagle Medium, high glucose, GlutaMAX Supplement (DMEM + GlutaMAX, Gibco) supplemented with 10% fetal bovine serum (FBS, Gibco), 100 units/mL of penicillin, 100 μg/mL of streptomycin (Gibco) until 90–95% confluence in T-75 flasks prior to subculture or use. 3T3-L1 cells were grown at 37 °C and 5% $CO_2$ in DMEM, high glucose, GlutaMAX, pyruvate (Gibco) supplemented with 10% bovine calf serum (BCS, Gibco) and 100 units/mL of penicillin, 100 μg/mL of streptomycin (Gibco) until 70% confluence in T-75 flasks prior to subculture or use.

### Protein expression and purification
Human PPARγ LBD protein, residues 203−477 (isoform 1 numbering), was expressed as a TEV-cleavable N-terminal hexa-his-tag fusion protein (6xHis-PPARγ LBD). Expression was performed in BL21(DE3) *Escherichia coli* cells in either autoinduction ZY media (unlabeled protein) or using M9 minimal media supplemented with ¹⁵N ammonium chloride (for NMR studies). For autoinduction, cells were grown at 37 °C for 5 h, 30 °C for 1 h, and 18 °C for 16 h before harvesting by centrifugation (4000 g, 30 min). For minimal media, cells were grown until the $OD_{600}$ was 0.6 before adding 0.5 mM (final concentration) isopropyl β-D-thiogalactoside (IPTG) and incubating at 18 °C for 16 h before harvesting by centrifugation (4000 g, 30 min). Cells were resuspended in lysis buffer (50 mM potassium phosphate (pH 7.4), 500 mM KCl, 10 mM imidazole) and lysed by sonication on ice. Cell lysate was clarified by centrifugation (20,000 g, 30 min) and filtration (0.2 μm). For His-tagged PPARγ-LBD, the protein was purified using Ni-NTA affinity chromatography followed by size exclusion chromatography (Superdex 75) on an AKTA pure in assay buffer (20 mM potassium phosphate, pH 7.4, 50 mM KCl, 0.5 mM EDTA). For NMR and crystallography, the His-tag was cleaved with TEV protease in dialysis buffer (20 mM potassium phosphate pH 7.4, 200 mM KCl) overnight at 4 °C. The cleaved protein was reloaded on the Ni-NTA column, the flow through collected, and further purified by size exclusion chromatography (Superdex 75) in assay buffer (20 mM potassium phosphate, pH 7.4, 50 mM KCl, 0.5 mM EDTA). Protein was confirmed to be >95% pure by SDS-PAGE. Purified samples were stored at −80 °C.

### Time-resolved fluorescence resonance energy transfer (TR-FRET) coregulator interaction assays
Assays were performed in black 384-well plates (Greiner) with 23 μL final well volume. For the coregulator recruitment assay, each well containing 4 nM 6xHis-PPARγ LBD, 1 nM LanthaScreen Elite Tb-anti-His Antibody (Thermo Fisher; dilution/amount used per product guidelines), and 400 nM FITC-labeled NCoR1 ID2 or MED1 ID2 peptide in a buffer containing 20 mM potassium phosphate (pH 7.4), 50 mM potassium chloride, 5 mM TCEP, 0.005% Tween 20. Ligands were added as a single concentration (10 μM) or in dose-response format to determine $EC_{50}/IC_{50}$ values. Compound stocks were prepared in DMSO, added to wells in triplicate, and plates were read using BioTek Synergy Neo multimode plate reader after incubation at 25 °C for 1 h. The Tb donor was excited at 340 nm; the Tb donor emission was measured at 495 nm, and the acceptor FITC emission was measured at 520 nm. Data were plotted using GraphPad Prism as TR-FRET ratio (520 nm/495 nm) vs. ligand concentration; fit to a three-parameter sigmoidal dose-response equation to obtain potency values; and are representative of two or more independent experiments.

### Fluorescence polarization coregulator interaction assays
Assays were performed using 6xHis-PPARγ LBD preincubated with or without 2 molar equivalents of compound at 4 °C overnight and buffer exchanged via centrifugal concentration using Amicon Ultra centrifugal filters to remove excess ligand. Protein samples were serially diluted into a buffer containing 20 mM potassium phosphate (pH 8), 50 mM potassium chloride, 5 mM TCEP, 0.5 mM EDTA, and 0.01% Tween 20 and plated with 180 nM FITC-labeled NCoR1 ID2 or TRAP220/MED1 ID2 peptide in black 384-well plates (Greiner). The plate was incubated at 25 °C for 1 h, and FP was measured on a BioTek Synergy Neo multimode plate reader at 485 nm emission and 528 nm excitation wavelengths. Data were plotted using GraphPad Prism as FP signal in millipolarization units vs. protein concentration (*n* = 3 biological replicates); fit to a one site−total binding equation using a consistent, fixed Bmax value determined from a fit of the high-affinity interactions as binding for some conditions did not saturate at the highest protein concentration used (45 μM); and are representative of two or more independent experiments.

### Transcriptional reporter assays
HEK293T cells were cultured in Dulbecco's minimal essential medium (DMEM, Gibco) supplemented with 10% fetal bovine serum (FBS) and 50 units mL⁻¹ of penicillin, streptomycin, and glutamine. Cells were grown to 90% confluency in T-75 flasks; from this, 2 million cells were seeded in a 10-cm cell culture dish for transfection using X-tremegene 9 (Roche) and Opti-MEM (Gibco) with full-length human PPARγ isoform 2 expression plasmid (4 μg), and a luciferase reporter plasmid containing the three copies of the PPAR-binding DNA response element (PPRE) sequence (3xPPRE-luciferase; 4 μg). After an 18-h incubation, cells were transferred to white 384-well cell culture plates (Thermo Fisher Scientific) at 10,000 cells/well in 20 μL total volume/well. After a 4 h incubation, cells were treated in quadruplicate with 20 μL of either vehicle control (1.5% DMSO in DMEM media) or 5 μM ligand, or in dose-response format to determine $EC_{50}/IC_{50}$ values, where each ligand treated condition had separate control wells to account for plate location-based artifacts. After a final 18-h incubation, cells were harvested with 20 μL Britelite Plus (PerkinElmer), and luminescence was measured on a BioTek Synergy Neo multimode plate reader. Data were plotted in GraphPad Prism as luciferase activity vs. ligand concentration (*n* = 4 biological replicates); fit to a three-parameter sigmoidal dose-response equation to obtain potency values; and are representative of two or more independent experiments.

## Gene expression analysis

3T3-L1 cells were cultured in DMEM medium supplemented with 10% FBS and 50 units $mL^{-1}$ of penicillin, streptomycin, and glutamine. Cells were grown to 70% confluency and then seeded in 12-well dishes at 50,000 cells per well and incubated overnight at 37 °C, 5% CO2. The following day, cells were treated with media supplemented with 0.5 mM 3-iso-butyl-1-methylxanthine, 1 µM dexamethasone, and 877 nM insulin. Following 2 days of incubation, cells were treated with 10 µM compound in media supplemented with 877 nM insulin for 24 h. RNA was extracted using quick-RNA MiniPrep Kit (Zymo) and used to generate complementary DNA using qScript cDNA synthesis kit (Quantabio). Expression levels of the PPARγ target gene *aP2/FABP4* (forward primer: 5′-AAGGTGAAGAGCATCATAACCCT-3′) and the housekeeping gene *TBP* (forward primer: 5′-ACCCTTCACCAAT-GACTCCTATG-3′) used for normalization was measured using Applied Biosystems 7500 Real-Time PCR system. Relative gene expression was calculated via the ddCt method using Applied Biosystems Relative Quantitation Analysis Module Software, which reported values as mean with upper and lower limits. Data were plotted in GraphPad Prism and are representative of two or more independent experiments. Three of the compounds in our ligand series—GW9662, **5** (SR36706), and **11** (SR33486)—and the control agonist rosiglitazone were not available or at the time these experiments were performed and left out of this analysis.

## NMR spectroscopy

Two-dimensional $[^{1}H,^{15}N]$-TROSY-HSQC NMR data (Bruker pulse sequence = trosyf3gpphsi19) of $^{15}N$-labeled PPARγ LBD (200 µM) were acquired at 298 K on a Bruker 700 MHz NMR instrument equipped with a QCI cryoprobe in NMR buffer (50 mM potassium phosphate, 20 mM potassium chloride, 1 mM TCEP, pH 7.4, 10% D2O) with ligands preincubated overnight at 4 °C with 2 molar equivalents and buffer exchanged via centrifugal concentration using Amicon Ultra centrifugal filters to remove excess ligand. Data were collected using Topspin 3.0 (Bruker Biospin) and processed/analyzed using NMRFx (version 11.4)[42]. NMR chemical shift assignments transferred from rosiglitazone-bound PPARγ LBD[14] to T0070907- and GW9662-bound states[22,24] were used in this study for well-resolved residues with conversed NMR peak positions to the previous ligand-bound forms using the minimum chemical shift perturbation procedure[43]. Population-weighted $^{1}H$ NMR chemical shift analysis was performed focusing on the NMR peaks observed for Gly399 in each ligand-bound NMR spectrum. Peak volumes for well-resolved peaks (e.g., one peak = one state/conformation, two peaks = two states/conformations in slow exchange) were calculated using an elliptical peak fitting algorithm. Population weighted average $^{1}H$ NMR chemical shift values were calculated with Python using Jupyter Notebook using NumPy and Pandas packages using an equation ("lambda x: np.average(x.H1_P, weights=x.weighted_vol)") with two inputs: peak volumes estimated the relative population sizes (weighted_vol) of each state, and the $^{1}H$ NMR chemical shift values of each state (H1_P).

## Crystallization and structure determination

Two molar equivalents of compounds were added to purified PPARγ-LBD (1:2 protein/compound ratio) and incubated at 4 °C overnight, followed by the addition of 5 molar equivalents of appropriate peptides (1:5 protein/peptide ratio) and a final incubation at 4 °C overnight. Protein complexes were purified by gel filtration (Superdex 75) on an AKTA pure, in assay buffer (20 mM potassium phosphate, pH 7.4, 50 mM KCl, 0.5 mM EDTA), and concentrated to 10 mg/mL. Crystals of the protein complex were obtained by sitting-drop vapor diffusion against 50 µL of reservoir solution (100 mM MES [pH 6.5], 200 mM ammonium sulfate, 30% PEG 8000) at a 1:1 protein/reservoir solution ratio using 96-well crystallization plates at 22 °C. Crystals were transferred to cryoprotectant (reservoir solution plus 10% ethylene glycol)

and flash-frozen in liquid nitrogen prior to data collection at the SLAC National Accelerator Laboratory/Stanford Synchrotron Radiation Lightsource (SSRL) beamline 12-2. Data were processed with XDS and scaled with Aimless. Structures were solved by molecular replacement using Phaser in the Phenix software package, using the crystal structure of PPARγ bound to T0070907 and NCoR1 ID2 peptide (PDB code: 6ONI) as the search model. Structures were built with iterative rounds of manual model rebuilding in COOT followed by refinement using phenix.refine. Pairwise structural alignment and rmsd calculations were performed via the RCSB webserver (https://www.rcsb.org/alignment/) using the jFATCAT rigid structural alignment algorithm.

## Density functional theory (DFT) quantum mechanical (QM) binding free energy calculations

We evaluated the binding free energies of ligands through the QM cluster method. For each PPARγ-ligand complex, we curated a cluster structure comprising the ligand and binding site residues (i.e., His323, His449, Tyr473, Cys285, Gln286, Tyr327, Met364, Lys367) pi-stacking aromatic triad interaction network and nearby residues based on the crystal structure of PPARγ LBD bound to T0070907 inverse agonist and NCoR1 corepressor peptide (PDB 6ONI)[24]. The cluster is further converted into the complex, ligand, and binding site models for subsequent binding free energy calculation. Using Gaussian16[44], these structural models were optimized with the PBE0-D3/def2-SVP method under fixed Cα and Cβ coordinates, followed by a single point energy correction with the PBE0-D3/def2-TZVP method. Each optimized structure underwent normal mode analysis with the PBE0-D3/def2-SVP method to validate that it is a stationary point on the potential energy surface. The analysis also informs the free energy correction value, which was further corrected using the quasi-harmonic method[45] and then added to the single point energy to give the standard Gibbs free energy of the structure. Eventually, the binding free energy was calculated by subtracting the free energy of the complex from the free energy sum of the ligand and binding site.

## Statistical analysis

Correlation data plotting and analysis of Spearman (s) correlation coefficients and two-sided alternative hypothesis *p* value testing (without adjustments for multiple comparisons) of the biochemical and cellular ligand profiling data were performed in Python using Jupyter Notebook using several libraries including Seaborn, Matplotlib, Numpy, and Scipy. Reported two-sided *p* values (*p*) represent the probability that the absolute value of the Spearman or Pearson coefficient of random (*x*, *y*) value drawn from the population with zero correlation would be greater than or equal to abs(*r* or *s*), according to the Scipy stats documentation. Data in figure legends are reported as (*n* = *X* biological replicates, mean ± s.e.m.), for example.

## Reporting summary

Further information on research design is available in the Nature Portfolio Reporting Summary linked to this article.

# Data availability

Crystal structures generated in this study have been deposited in the Protein Data Bank (PDB) under accession codes 8FHE, 8FHG, 8FHF, 8FKC, 8FKD, 8FKE, 8FKF and 8FKG. Other crystal structures previously deposited in the PDB used in this study include 3B0R, 6C1I, 6ONI and 6ONJ. Previous published NMR chemical shift assignments used in this study include BMRB accession codes 17975 and 50000. Source data and analysis scripts are available with this paper as a Source Data file underlying Figs. 1d, e, 3, 4 and 6b; Supplementary Figs. 2 and 3; input and output files for structural models used in the DFT-QM calculations; and Jupyter notebook python scripts used to calculate NMR population-weighted average $^{1}H$ chemical shift values (Fig. 6b) and correlation analyses (Figs. 4 and 6b; Supplementary Figs. 2 and 3).

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

## Acknowledgements

This work was supported in part by the National Institutes of Health (NIH) grants R01DK124870 (D.J.K.) from the National Institute of Diabetes and Digestive and Kidney Diseases (NIDDK) and R35GM146982 (Z.J.Y.) from the National Institute of General Medical Sciences (NIGMS). Use of the Stanford Synchrotron Radiation Lightsource, SLAC National Accelerator Laboratory, is supported by the U.S. Department of Energy, Office of Science, Office of Basic Energy Sciences under Contract No. DE-AC02-76SF00515. The SSRL Structural Molecular Biology Program is supported by the DOE Office of Biological and Environmental Research and by the NIH NIGMS grant P30GM133894. The contents of this publication are solely the responsibility of the authors and do not necessarily represent the official views of NIDDK, NIGMS, NIH, or DOE.

## Author contributions

B.S.M. and D.J.K. conceived and designed the research. D.Z., Y.H. and T.M.K. synthesized and characterized compounds. B.S.M. and J.S. expressed and purified protein and solved crystal structures. B.S.M. performed biochemical, cellular, and NMR studies. Q.Z. and Z.J.Y. performed DFT-QM calculations. All authors analyzed data. D.K. supervised the research and wrote the manuscript along with B.S. and input from all authors who approved the final version.

## Competing interests

The authors declare no competing interests.
