## [Transparent Peer Review file · Nature Communications]

Ligand efficacy shifts a nuclear receptor conformational ensemble between transcriptionally active and repressive states

Corresponding Author: Professor Douglas Kojetin

This manuscript has been previously reviewed at another journal that is not operating a transparent peer review scheme. The manuscript was considered suitable for publication without further review at Nature Communications.

Version 0:

Reviewer comments:

Reviewer #1

(Remarks to the Author)

The manuscript by Brian MacTavish et al. reported a structure-function study of a series of ligands targeting the PPAR γ receptor. Compounds showed functional profile ranging from inverse agonist to agonist. Coregulators (NCOR1 and MED1) binding to PPAR γ complexes were measured as well as reporter assay and gene activation assay. NMR analysis focused on some PPAR γ specific amino acid peaks were analyzed for all complexes and for some of the ligands, crystal structures were determined.

The novelty of the manuscript mainly relies on the analysis of a large number of ligands. The finding of a PPAR γ conformational ensemble shift between 2 states were already described by the corresponding author (Shang et al 2020, 2021, Frkic,..) as the adoption of 2 states in ligand conformations.

More importantly, the analyses of the data are very superficial.

-More details of the chemical modifications should be provided.

-Whereas good correlations between the assays were presented, some outliers are also observed and not discussed. For example, ligand 20 that binds with good affinity NCOR, is as active as compounds 18,19. Why some differences in the reporter assay and transcription assay for some ligands? Compounds 17, 19, 20 showed increased transcription, but binding to MED1 is 1-2 orders of magnitude lower to the previously measured affinity of agonist ligands by the corresponding author.

-The x-ray structures should be more discussed for all complexes. What is the correlation of increased binding to NCoR for ligand 11? DeltaG analyses do not provide much information.

-Interpretation of NMR peaks is more complex than the conclusion of a shift between active and repressive single conformations. For some ligands, peaks are double, and not positioned similarly compared to the control ligands.

-Standard errors on the experimental data should be given in the source data file.

- A control agonist ligand should have been included in the analysis.

-HPLC traces profiles of the new synthesized ligands should be included.

Reviewer #2

(Remarks to the Author)

The work entitled: "Ligand efficacy shifts a nuclear receptor conformational ensemble between transcriptionally active and repressive states" describe the authors efforts to create a framework to design and rationalize the improve activity of Peroxisome proliferator-activated receptor gamma (PPAR γ). The manuscripts is well written and is of interest for the readership of Nature Communications. I suggest to publish it in Nat. Commun. after the following comments have been addressed. I have only two comments on the DFT calculations part:

1. The authors performed DFT calculations to predict interaction energies. How the residues were chosen for including in

the QM calculations? Are the selection based on a distance cutoff?

2. The calculations should be performed including the effect of the protein environment. Using QM/MM simulations this can be achieved. The importance of considering other effects in the binding of the substrate is clear in the results on the pyridyl substituted analogs where there is no good correlation with the experimentally observed results.

Reviewer #3

(Remarks to the Author)

This research will be significant to the field of PPAR γ and other nuclear receptors, but will also be of significant interest in other areas of structural biology and protein function. The role of conformational ensembles in protein function is now generally recognized, but there are few studies that make this role as clear as that done in the research described here. Structural studies still often focus on a static description of structures. The powerful combination of X-ray crystallography, NMR spectroscopy, and biochemical assays utilized here shows how these complementary methods can provide a more complete understanding of the structural ensembles. Previous studies of PPAR γ structural biology have generally focussed on comparison of x-ray and/or NMR studies comparing apo-PPAR γ and one or a few ligands. Here, these methods are used to probe a much larger number (20) of compounds that form a graded range of activation and repression. These compounds, with only minor chemical changes, provide a unique probe of PPAR γ function. In particular they give access to an understanding of the mechanism of inverse agonism, not well seen with previous compounds.

I strongly support publication of this manuscript, with the few minor changes and questions listed below.

One minor error, the legend to Figure 4 indicates that the data are colored according to compound numbers in Figure 1. I think this should be Figure 2.

One suggestion, In figure 3 the vertical axis labels are all the same. The figure would be clearer, with more space given to the data if only one axis (the left most) was labeled with the compound name. And then possibly, faint horizontal lines could be used to connect connect the dots across the four sections.

It would be useful to mention what method was used for the buffer exchange to remove excess ligand before NMR spectroscopy.

The clash-score for structure in file "507581_0_related_ms_9035270_sd6ym7" seems significantly worse than that for the other structures. Does this have any impact on interpretation of this structure.

Reviewer #4

(Remarks to the Author)

Kojetin and co-workers report a mechanistic study on PPAR γ ligand bias, i.e., the structural determinants and molecular mechanisms of PPAR γ activation and repression using structural and NMR analysis. A similar though less detailed studied has been performed previously for FXR (10.1038/s41467-019-10853-2). The authors capitalize on three commercially available inverse PPAR γ agonists for initial studies and subsequently present a tailored library of analogues spanning agonism to inverse agonism as a tool to explore the molecular differences between activation and repression. This choice of structurally related tool compounds for comparative structural analysis and for establishing structure-conformation relationships is a strength. Although PPAR γ is among the best studied nuclear receptors, the insights from this study offer substantial mechanistic advance for the field. The study is well designed, the manuscript is well written, and the conclusions are sound. However, the following points should be addressed before publication.

The manuscript lacks information on the potencies of the newly reported PPAR γ modulators and on the concentrations used for the comparative mechanistic studies in Fig. 1e and Fig. 3. This missing information should be added along with explanation how the concentrations were chosen, at best based on potency or affinity data for all studied compounds from a uniform test system.

Fig. 6a would benefit from data for a TZD full agonist for comparison.

The NMR studies focus on Gly399 to observe conformational differences between active and repressed states. This is well explained but a structural figure showing the location of Gly399 in the PPAR γ LBD in complex with coactivator/-repressor might help understanding.

The relevance of the ligand bound active conformations of the PPAR γ LBD (Fig. 1h) obtained by soaking may be questioned especially considering that ZINC5672437 almost fully favors the repressed conformation in NMR. This raises the question what effect the addition of a coactivator would have in the NMR experiments with and without NCOR in Fig. 1f and 6a?

Explanation why few compounds were not used in the 3T3-L1 differentiation and gene expression experiment should be added.

Although the series of T0070907 analogues is well designed it is surprising that polar 4-substituents like hydroxy or dimethylamino to dis-/replace the bound water are missing.

Minor points

The color code in Fig. 5 should be explained in the caption.

There are more recent examples of PPAR γ ligands that can bind simultaneously with GW9662 (e.g., 10.1016/j.chembiol.2021.04.019).

Version 1:

Reviewer comments:

Reviewer #1

(Remarks to the Author)

The authors have adequately addressed my suggestions and questions. The revised version is clearly improved. The manuscript should be ready for publication in my opinion

Reviewer #2

(Remarks to the Author)

The authors have addressed my concerns and explain clearly their reasoning to not perform the additional QM/MM calculations. I believe the paper can be published in its current form.

Reviewer #3

(Remarks to the Author)

The revised manuscript should be accepted for publication

Reviewer #4

(Remarks to the Author)

The authors have convincingly addressed all my comments and present a very nice and timely study that advances research on PPAR γ and molecular understanding of nuclear receptors in general. The manuscript appears suitable for publication in Nature Communications in its present state.

We thank the reviewers for their time and constructive comments, and we hope that the points outlined below are adequately addressed in our revised manuscript. Please note that all source data are now located in a zipped folder (Source Data.zip) per journal requirements.

REVIEWER COMMENTS

Reviewer #1 (Remarks to the Author):

The manuscript by Brian MacTavish et al. reported a structure-function study of a series of ligands targeting the PPAR γ receptor. Compounds showed functional profile ranging from inverse agonist to agonist. Coregulators (NCOR1 and MED1) binding to PPAR γ complexes were measured as well as reporter assay and gene activation assay. NMR analysis focused on some PPAR γ specific amino acid peaks were analyzed for all complexes and for some of the ligands, crystal structures were determined.

The novelty of the manuscript mainly relies on the analysis of a large number of ligands. The finding of a PPAR γ conformational ensemble shift between 2 states were already described by the corresponding author (Shang et al 2020, 2021, Frkic,..) as the adoption of 2 states in ligand conformations.

More importantly, the analyses of the data are very superficial.

- More details of the chemical modifications should be provided.

The original focus we wished to convey in this manuscript was to highlight how NMR spectroscopy can be used to determine how ligand modifications influence a nuclear receptor's conformation-activity relationship (CAR) to describe the activity of a ligand set, which is distinct from traditional ligand structure-activity relationship (SAR) analysis that focuses on ligand modifications and changes in functional activity. In our revised manuscript, we added some details concerning the chemical modifications and their associated effects on PPAR γ function, though we also state where additional SAR on a larger ligand series would be needed to confirm the observations we describe.

- Whereas good correlations between the assays were presented, some outliers are also observed and not discussed. For example, ligand 20 that binds with good affinity NCOR, is as active as compounds 18,19. Why some differences in the reporter assay and transcription assay for some ligands? Compounds 17, 19, 20 showed increased transcription, but binding to MED1 is 1-2 orders of magnitude lower to the previously measured affinity of agonist ligands by the corresponding author.

There are indeed a few ligands with activities that deviate from a perfect correlation, in particular when translating (1) biochemical assay data using with purified reconstituted PPAR γ LBD protein/coregulator peptides to (2) cellular transcription (with full-length PPAR γ) and gene expression (endogenous PPAR γ in a chromatin-bound context) data in a more complex endogenous environment with many different coregulator proteins that can contribute to the activities. It is possible these features may contribute to the difference in activities when comparing compounds 20 vs. 18 and 19, where there appears to be a strong correlation between biochemical TR-FRET endpoint values (Figure 3A) and coregulator binding affinity via FP (Figure 3B). However, the correlation to cellular data is still relatively strong but nonetheless lower than comparing among the biochemical data only.

Related to the influence of compounds 17, 19, and 20 on MED1 binding affinity relative to other agonist ligands we have studied (e.g., Shang et al, PNAS 2019 29;116(44):22179-22188) — the agonists within the ligand set in this current manuscript display comparatively low efficacy (low partial agonism) compared to most of the TZD ligands we studied in the PNAS manuscript, which ranged from partial agonism (a few ligands) to full agonism (most ligands). For comparison, we added the full agonist rosiglitazone, which was used as a control in our the functional profiling assays, in Figure 3 and NMR analysis in Figure 6a. Notably, however, two TZDs (CAY10638 and CAY10405) in our previous study that displayed the lowest efficacy (partial agonism) result in a MED1/TRAP220 binding affinity ($\log M = \sim -5.6$ to -5.9) in the same range as the compounds 17-20 in this current study ($\log M = \sim -5.8$ to -5.9), which is consistent with the partial agonism transcription function of compounds 17-20.

We also added two new functional ligand profiling data analyses (Figure 3c) — termed “coregulator bias”, which reports on the difference between the TR-FRET and FP value (NCoR1 value – MED1 value); that is, how much NCoR1 interaction is strengthened relative to weakening of MED1. These new analyses, which are also added to the correlation plot analysis in Figure 4, give a different perspective in understanding how transcriptional efficacy relates to the magnitude of change in the NCoR1 and MED1 biochemical assays. The coregulator bias analysis shows just as strong, if not stronger, correlation compared to the individual TR-FRET or FP correlations with NCoR1 and MED1 peptide. For example, spearman correlation coefficients extracted from these updated analyses shown in Supplementary Figure 1 are shown below:

	NCoR1 FP	MED1 TR- FRET	MED1 FP	TR- FRET bias	FP bias	Transcription	aP2 expression	NMR
NCoR1 TR-FRET	-0.96	-0.90	0.93	0.99	-0.96	-0.97	0.86	0.86

- The x-ray structures should be more discussed for all complexes. What is the correlation of increased binding to NCoR for ligand 11? DeltaG analyses do not provide much information.

We added additional descriptions of the crystal structures to the results section. As to why compound 11 results in increased binding of NCoR1, we reason it is likely due to several factors including (1) replacement of the water that bridges interaction between T0070907 and the His323 side chain and (2) retention of the polar pyridyl ring interaction with the aromatic triad. However, compound 11 shows lower (graded) repressive efficacy in the biochemical and cellular assays. We posit this may be due to the pyridyl nitrogen in the 1-substituted position, which in our crystal structure points towards the Gln286 side chain (or Tyr327 in the flipped ligand orientation, as the electron density underlying our x-ray crystal structure cannot discriminate between either orientation). Additional SAR analysis, which we plan to do in follow-on studies, may uncover the influence of placing a pyridyl nitrogen in the 1-substituted position vs. 2- and 3-substituted positions, for example.

- Interpretation of NMR peaks is more complex than the conclusion of a shift between active and repressive single conformations. For some ligands, peaks are double, and not positioned similarly compared to the control ligands.

As the reviewer indicates, some of the compounds when bound to PPAR γ LBD produce multiple ligand-bound peaks for the same residue. There are several possible interpretations for the appearance of multiple peaks: (1) there could be exchange between multiple ligand-bound (sub)conformations; (2) the aromatic ring could flip orientations within the orthosteric ligand-binding pocket as part of the exchange mechanism; and (3) the typically weaker (lesser) intensity peak nearer the position of the apo/ligand-free conformation represents an exchange peak to the “other” conformation—using compound 2 as an example, the high intensity peak is the repressive conformation and the lower intensity peak denotes exchange to an active conformation but the kinetics of the exchange is different than compound 4, which also shows two peaks but the weaker peak is further shifted towards the active state. It is important to note that when multiple peaks appear they fall along a diagonal line connecting the active and repressive conformations. Thus, if a ligand-bound state shows multiple peaks, the NMR data suggest that, using compound 19 as an example, the peak shifted more up and to the right (lower ^{15}N and ^1H chemical shift values) represents a more active conformation.

- Standard errors on the experimental data should be given in the source data file.

We updated the Source Data 1 file to contain the actual experimental data plotted in the Figures including experimental replicates, standard, or fitted errors where applicable.

- A control agonist ligand should have been included in the analysis.

We added data for a control agonist (rosiglitazone) when it was included in the ligand profiling assays in Figure 3 (TR-FRET, FP, and luciferase reporter assay; but not the 3T3-L1 experiment) and Figure 6a. We did not include rosiglitazone in the correlation analysis since this is focused on describing the activity of the ligand series; however, we note that including adding rosiglitazone to the correlation analysis does not really change the spearman correlation coefficients reported in Figure 4 and Supplementary Figure 1.

- HPLC traces profiles of the new synthesized ligands should be included.

The author guidelines for chemical characterization and purify at Nature Communications (<https://www.nature.com/ncomms/submit/chemical-characterisation>) and other Nature branded journals indicates that HPLC is not required via the phrasing “...may be used..”: “For most organic and organometallic compounds, purity may be demonstrated by high-field ^1H NMR or ^{13}C NMR data, although elemental analysis ($\pm 0.4\%$) is encouraged for small molecules. Quantitative analytical methods including chromatographic (GC, HPLC, etc.) or electrophoretic analyses may be used to demonstrate purity for small molecules and polymeric materials.” In our original submission, we included a Supporting Information document labeled “Supplemental File 1” that contains ^1H NMR (chemical shifts and spectra), ^{13}C NMR (chemical shifts and spectra) and mass spectrometry data, which meets the standard journal requirements. The editor reached out to this reviewer and confirmed that the ^1H NMR, ^{13}C NMR, and mass spectrometry mass analysis we included satisfies the Nature author guidelines (thank you).

Reviewer #2 (Remarks to the Author):

The work entitled: “Ligand efficacy shifts a nuclear receptor conformational ensemble between transcriptionally active and repressive states” describe the authors efforts to create a framework to design and rationalize the improve activity of Peroxisome proliferator-activated receptor gamma (PPAR γ). The manuscripts is well written and is of interest for the readership of Nature Communications. I suggest to publish it in Nat. Commun. after the following comments have been addressed. I have only two comments on the DFT calculations part:

1. The authors performed DFT calculations to predict interaction energies. How the residues were chosen for including in the QM calculations? Are the selection based on a distance cutoff?

We chose the residues based on the pi-stacking aromatic triad interaction network and nearby residues reported by Shang et al. 2020 (ref. 24) that reported the crystal structure of PPAR γ LBD bound to T0070907 inverse agonist and NCoR1 corepressor peptide (PDB 6ONI). We have updated the methods section to reflect this.

2. The calculations should be performed including the effect of the protein environment. Using QM/MM simulations this can be achieved. The importance of considering other effects in the binding of the substrate is clear in the results on the pyridyl substituted analogs where there is no good correlation with the experimentally observed results.

We thank the reviewer for the suggestion. We want to clarify that the goal of DFT (QM cluster) calculations is not to assess the effect of the compounds on substrate (peptide) binding affinity, but to explore the relationship between the strength of the compound interaction with the aromatic triad and nearby residues and the dynamic LBD exchange mechanism caused by the compounds. We focused on the compound interaction network with the repressive conformation of the protein and hypothesized that the stronger the interaction of the inverse agonists with the residues in the interaction network, the more preferred the repressive conformation will be in the allosteric exchange mechanism. Adding more protein environment to the model by QM/MM offers limited help in understanding this behavior as it focuses on complementing the electrostatics environment of the protein-compound interaction. We think an important factor missing in these computational studies is a focus on the allosteric exchange mechanism, which is complex and would require very long time-scale simulations and computational resources, given our NMR data that show the compound-bound protein exchanges between different conformations in slow exchange on the NMR time scale indicating the exchange occurs on the millisecond to seconds time scale, which is too long of a time scale for QM/MM calculations (nanosecond time scale).

Reviewer #3 (Remarks to the Author):

This research will be significant to the field of PPAR γ and other nuclear receptors, but will also be of significant interest in other areas of structural biology and protein function. The role of conformational ensembles in protein function is now generally recognized, but there are few studies that make this role as clear as that done in the research described here. Structural studies still often focus on a static description of structures. The powerful combination of X-ray crystallography, NMR spectroscopy, and biochemical assays utilized here shows how these complementary methods can provide a more complete understanding of the structural ensembles. Previous studies of PPAR γ structural biology have generally focussed on comparison of x-ray and/or

NMR studies comparing apo-PPAR γ and one or a few ligands. Here, these methods are used to probe a much larger number (20) of compounds that form a graded range of activation and repression. These compounds, with only minor chemical changes, provide a unique probe of PPAR γ function. In particular they give access to an understanding of the mechanism of inverse agonism, not well seen with previous compounds.

I strongly support publication of this manuscript, with the few minor changes and questions listed below.

One minor error, the legend to Figure 4 indicates that the data are colored according to compound numbers in Figure 1. I think this should be Figure 2.

We made this correction in the legend of Figure 4 and Figure 6.

One suggestion, In figure 3 the vertical axis labels are all the same. The figure would be clearer, with more space given to the data if only one axis (the left most) was labeled with the compound name. And then possibly, faint horizontal lines could be used to connect the dots across the four sections.

We revised the appearance of Figure 3 according to this suggestion. We now only call out compound names on the left side of the figure (in panel A) and reformatted the plots. We placed a dotted horizontal line across the figure panels for the two reference parent compounds (T0070907 and GW9662), which we think gives enough reference such that reads will be able to connect the dots across the five figure panels (previously four, but we added a new panel E). We agree makes the figure cleaner and clearer for readers.

It would be useful to mention what method was used for the buffer exchange to remove excess ligand before NMR spectroscopy.

We added this detail (buffer exchange via centrifugal concentration using Amicon Ultra centrifugal filters) to the NMR spectroscopy methods section.

The clash-score for structure in file “507581_0_related_ms_9035270_sd6ym7” seems significantly worse than that for the other structures. Does this have any impact on interpretation of this structure.

The clash score is pretty good relative to all structures in the PDB (upper ~25%) including structures at similar resolution (upper ~33%). There is no significant impact on the interpretation of the structure, in particular since there are no clashes around the ligand.

Reviewer #4 (Remarks to the Author):

Kojetin and co-workers report a mechanistic study on PPAR γ ligand bias, i.e., the structural determinants and molecular mechanisms of PPAR γ activation and repression using structural and NMR analysis. A similar though less detailed studied has been performed previously for FXR (10.1038/s41467-019-10853-2). The authors capitalize on three commercially available inverse PPAR γ agonists for initial studies and subsequently present a tailored library of analogues spanning agonism to inverse agonism as a tool to explore the molecular differences between activation and repression. This choice of structurally related tool compounds for comparative structural analysis

and for establishing structure-conformation relationships is a strength. Although PPAR γ is among the best studied nuclear receptors, the insights from this study offer substantial mechanistic advance for the field. The study is well designed, the manuscript is well written, and the conclusions are sound. However, the following points should be addressed before publication.

The manuscript lacks information on the potencies of the newly reported PPAR γ modulators and on the concentrations used for the comparative mechanistic studies in Fig. 1e and Fig. 3. This missing information should be added along with explanation how the concentrations were chosen, at best based on potency or affinity data for all studied compounds from a uniform test system.

We added potencies (IC₅₀ or EC₅₀ values) to the Source Data 1 Excel file from TR-FRET and transcriptional reporter assays collected during our iterative medicinal chemistry efforts where small groups of compounds were synthesized and tested, which informed additional changes and testing, and so on. Since those iterative studies were performed at different times throughout the project — and because we were most interested in comparing compound efficacy to protein conformation via NMR — the data shown in Figure 3 were collected using a single ligand concentration for all the compounds in the same experiment at the same time (10 μ M for TR-FRET, 5 μ M for transcription, 10 μ M for gene expression) or protein preincubated with compound (FP; 2x then buffer exchanged to remove excess ligand). We note that because these compounds bind via a covalent mechanism, the potency values will progressive left shift (become more potent) with time. We have included an explanation these concepts in the methods section “Ligands and peptides”.

Fig. 6a would benefit from data for a TZD full agonist for comparison.

We added data for a control agonist (rosiglitazone) when it was included in the ligand profiling assays in Figure 3 (TR-FRET, FP, and luciferase reporter assay; but not the 3T3-L1 experiment) and Figure 6a. We did not include rosiglitazone in the correlation analysis since this is focused on describing the activity of the ligand series; however, we note that including rosiglitazone does not significantly (or at all) change the spearman correlation coefficients reported in Figure 4 and Supplementary Figure 1.

The NMR studies focus on Gly399 to observe conformational differences between active and repressed states. This is well explained but a structural figure showing the location of Gly399 in the PPAR γ LBD in complex with coactivator/-repressor might help understanding.

In our revised Figures 1 and 5, we annotate the location of Gly399 on crystal structures of the PPAR γ LBD and call this out in the results section to help guide readers.

The relevance of the ligand bound active conformations of the PPAR γ LBD (Fig. 1h) obtained by soaking may be questioned especially considering that ZINC5672437 almost fully favors the repressed conformation in NMR. This raises the question what effect the addition of a coactivator would have in the NMR experiments with and without NCOR in Fig. 1f and 6a?

We crystallized ZINC5672437 in the active PPAR γ LBD conformation because although our NMR data show that this compound favors the repressive conformation, the active conformation is still populated; and we wanted to compare the active ZINC5672437 binding mode compared to the active T0070907 binding mode that we previously described in Shang et al. 2020 (ref. 24). As to the reviewer’s question, we although we did not run that specific experiment here, hypothesize that

since ZINC5672437 samples the active conformation that addition of coactivator would select the active state, similar to what we observed for T0070907 in Brust et al. 2018 (ref. 22).

Explanation why few compounds were not used in the 3T3-L1 differentiation and gene expression experiment should be added.

Unfortunately, we did not have three of the ligand series compounds or the control agonist (rosiglitazone) available when we ran the 3T3-L1 experiments; we added a sentence to the methods section to indicate this. However, CAR analysis of the ligand series extended to the 3T3-L1 data is still informative even if these compounds are not included. The advantage of our approach lies in the analysis of the ligand activities as a series/group instead of focusing on the activities of any individual ligand.

Although the series of T0070907 analogues is well designed it is surprising that polar 4-substituents like hydroxy or dimethylamino to dis-/replace the bound water are missing.

We originally synthesized three analogs with these substitutions (SR33066, SR33305, and SR33306) and tested them in some of the functional profiling assays. However, two of the analogs (SR33066 and SR33306) did not make the final list of compounds for our conformation-activity relationship (CAR) analysis on a smaller subset of ~20 compounds (SR33305 was included in this ligand series) that were selected for testing in all of the functional assays (Figure 3), correlation analysis of that data (Figure 4), and NMR analysis (Figure 6). In response to this comment, we now include all three compounds (SR33066, SR33305, and SR33306) in Figure 1 and updated our results section to describe their activities compared to GW9662 and T0070907. We hope this addition demonstrates a good faith effort to address this comment. We plan to perform SAR on a larger ligand series in the future to confirm these and several other observations we make in this revised manuscript focus primarily on CAR instead of SAR analysis.

Minor points

The color code in Fig. 5 should be explained in the caption.

We added an explanation to the Figure 5 caption.

There are more recent examples of PPAR γ ligands that can bind simultaneously with GW9662 (e.g., 10.1016/j.chembiol.2021.04.019).

Thank you for pointing this out. We have added this reference to the Introduction.